# Adverse Childhood Experiences and Health in Rural Areas of Riyadh Province in Saudi Arabia: A Cross-Sectional Study

**DOI:** 10.3390/healthcare10122502

**Published:** 2022-12-10

**Authors:** Fahad M. Alhowaymel, Atallah Alenezi

**Affiliations:** Department of Nursing, College of Applied Medical Sciences, Shaqra University, Shaqra 11911, Saudi Arabia

**Keywords:** rural, adverse childhood experiences, chronic diseases, risk behaviors, depression, anxiety

## Abstract

Adverse childhood experiences (ACEs) and their consequences are a worldwide problem. ACEs are primary stressors that have a long-term impact on the body and mind during development. They are linked to a variety of chronic illnesses in adults. Information about ACEs and health and risk behaviors is scarce among rural populations. The study aimed to examine ACEs, chronic diseases, and risk behaviors, as well as to explore the relationship between them and number of sociodemographic factors among adults living in rural areas in Riyadh Province, Saudi Arabia. A cross-sectional design and a convenience sampling method were utilized to collect information. A self-reported questionnaire, including an ACEs questionnaire as well as direct health and risk behaviors questions, was used. In total, 68.2% of the respondents reported at least one ACE, and 34.2% reported four or more ACEs. Emotional and physical abuse were the most reported forms. Hypertension and chronic respiratory disease were the most reported chronic diseases. Depression and anxiety were associated with ACEs, indicating that those who reported four or more ACEs were more likely to develop depression and anxiety. ACEs contribute to many negative health outcomes; thus, identifying the prevalence of ACEs among the rural population is essential for future health-related actions. It is also important that chronic diseases and risk behaviors be specifically identified among the rural population in order to prioritize these actions. Future research should further investigate ACEs and other determinants of health among the rural population, taking into consideration the inclusion of more diverse people, such as older adults and those from other rural areas.

## 1. Introduction

Adverse childhood experiences (ACEs) and their consequences are a worldwide problem [1]. ACEs are a major public health issue that affects all people, with long-term effects on general health as well as negative health, social, and financial consequences [2,3]. ACEs are commonly classified into one or more of three types of abuse. These are defined as childhood abuse and household disruptions experienced before the age of 18 that include exposure to physical, emotional, sexual, or substance abuse, as well as mental illness, imprisonment, separation, divorce, or adult violence [4,5,6]. According to a global epidemiologic study, childhood adversities are very widespread, with 38.8% of over 3700 adults across ten countries having experienced at least one ACE [7]. Indeed, ACEs, including childhood abuse and neglect, are identified as the most harmful sorts of adversity. Although ACEs are influenced by sociodemographic factors such as cultural, social, and environmental factors [8], they commonly occur across the world [7].

ACEs are primary stressors that have a long-term impact on the body and mind during development. They are linked to a variety of chronic illnesses in adults [9]. ACEs have been shown to cumulatively predict poor mental and physical outcomes, and the risk increases with greater exposure to ACEs [8,10]. There are long-standing consequences of ACEs for a variety of physiological systems, including the neuroendocrine, immune, and cardiovascular systems, as well as negative alterations to brain function and genetic aging [11,12]. Children who are exposed to a high amount of stress are more prone to adopting health-risk behaviors, such as smoking, consuming alcohol, or engaging in antisocial conduct. These high-risk habits result in poor adult health and an increased risk of physiological and psychological diseases [13,14]. The behavioral, biological, and psychosocial processes by which ACEs affect individual well-being are complicated and pervasive [15]. 

Several studies have linked ACEs to chronic diseases such as cancer; sexually transmitted diseases; frequent mental distress and depression; intimate partner violence; suicide attempts; health risk behaviors such as smoking, alcohol abuse, substance abuse, sexual risk-taking, and youth violence; as well as an increased risk of premature mortality before the age of 19 [16,17,18]. Indeed, research on ACEs and their consequences is increasingly growing worldwide; however, few studies have focused on rural populations. Among those studies, a study conducted across the United States of America found that rural children experienced more ACEs than urban children did. According to the researchers, poverty was a possible mitigation factor of the high experiences of ACEs among rural children [19]. Another study conducted in rural Pakistan found ACEs to be associated with higher levels of perceived stress, anxiety, and cortisol [20]. 

Rural areas are known for their low density and low economic resources, which reflect the presence of other resources, such as those related to health. Population in rural areas are mostly less educated and have lower incomes, in addition to having limited access to healthcare services, in developed countries such as the United Kingdom [21]. In Saudi Arabia, for example, services to screen and respond to childhood abuse are mainly provided in large cities, and not extended to rural areas [22]. Therefore, a lack of preventive measures and services may increase health problems among these populations. In addition to this, there is a scarcity of information about health among rural populations, and there is no evidence in the literature about ACEs in rural Saudi Arabia. Thus, the study aims to examine ACEs, chronic diseases, and risk behaviors, as well as to explore the relationship between them and number of sociodemographic factors in rural areas of Saudi Arabia. 

## 2. Materials and Methods

### 2.1. Study Design, Setting, and Participants 

A cross-sectional design was utilized to examine the relationship between ACEs, chronic diseases, risk behaviors, and sociodemographic factors in a rural population. The study was conducted in the northwestern region of Riyadh Province in Saudi Arabia. This region includes a number of rural areas that are located near or between suburban areas. Indeed, rural areas have lower populations than suburban and urban areas, most of them having less than five thousand people. Furthermore, rural populations have lower levels of education, lower income, and less access to healthcare services. Regarding Saudi Arabia, 20% of the total Saudi population lives in rural areas, and 8.5% of those from Riyadh Province lives in rural areas. Moreover, Riyadh Province includes approximately a hundred scattered rural areas [23]. The study included 338 participants. All of them lived in rural areas and were aged 18 years old or older. Excluded participants were those who did not live in rural areas during childhood, or who did not wish to participate. 

### 2.2. Measurement 

#### 2.2.1. Demographic Questions

The demographic questions in this study include six questions about age, gender, nationality, marital status, occupational status, and educational status. 

#### 2.2.2. Adverse Childhood Experience (ACE) Questionnaire 

The ACE Questionnaire is self-reported, contains 19 items, and is designed to measure exposure to ACEs retrospectively prior to age 18. The questionnaire was adapted by Kalmakis and colleagues [24] from the 10-item version that was developed by Dube and colleagues [25]. The questionnaire includes six items for physical, emotional, or sexual abuse, four items for neglect, and nine items for household dysfunction. All items in this questionnaire are binary (dichotomous), where the answers are “yes” or “no”. The score is summarized, resulting in a total score that ranges from 0 to 19. The following is an example of an item of the ACE Questionnaire: “Prior to your 18th birthday, did a parent or other adult in the household swear at you, insult you, put you down, or humiliate you?” 

An Arabic version of the ACE Questionnaire was used in this study. We translated the 19-item ACE Questionnaire using a forward–backward translation method (from the English to Arabic language and backward). Six bilingual experts in the field participated in the translation process. The final Arabic version of the ACE Questionnaire was piloted among 25 participants to assess its adaptability and accessibility. The results revealed that no changes were required after the pilot survey. The ACE Questionnaire had adequate test–retest reliability (r = 0.64, *p* < 0.001) [25], (r = 0.71, *p* < 0.001) [26]. The internal consistency reliability of the adapted 19-item ACE Questionnaire was acceptable (α = 0.835) [24]. In this study, the 19-item ACE Questionnaire—Arabic version had an acceptable internal consistency reliability (α = 0.844). 

#### 2.2.3. Health and Risk Behaviors Questions

The health and risk behavior questions included 15 items. Eleven questions were included to determine whether participants had been ever diagnosed with certain common diseases. For example, we asked participants if they have ever been diagnosed with diabetes, hypertension, depression, or anxiety. Four questions were added to measure common risk behaviors among adults. For example, we asked participants if they were current or previous smokers, and if they were regularly exercising. Similar studies in Saudi Arabia have used the same questions [22]. 

### 2.3. Data Collection

A non-probability convenience sampling method was used to recruit participants for this study at multiple sites in the northwestern region of Riyadh Province. The authors trained data collectors (research assistants) to participate in data collection processes. After completing training sessions, data collectors prepared the data collection scenes at public gathering areas, including small shopping centers and public parks. The scenes constituted of small portable rooms with partitioners to ensure privacy of the participants. Data collectors recruited participants using flyers and by providing a brief description of the study. When eligible participants agreed to participate, they entered the rooms and used smart portable tablets which were already prepared to fill out the questionnaires. Before proceeding to the main questionnaire, participants were required to answer the first question, which asked whether they agreed to participate in the study or not. If they answered with yes, they would move to the next question about their area of residence during childhood. If participants answered that they had lived in rural areas, they would continue to the main questionnaire; otherwise, the questionnaire would close. Then, eligible participants filled out their demographic information and proceeded to answering the main questions. The data collection phase took place during April 2022.

### 2.4. Ethical Consideration

The design and data collection for this study was approved by the Standing Committee of Research Ethics at Shaqra University (Ref.#: ERC SU 20210056). To maintain full privacy and confidentiality, all participants were made aware of the study’s aims. They were also aware of their right to withdraw from participating in the study at any time, without any consequences. Additionally, data collectors did not obtain any identifiers or personal information for personal privacy and to protect participants’ information. Study data were stored on the investigator’s personal computer. 

### 2.5. Data Analysis 

Analysis was conducted using the IBM-SPSS software version 28. Descriptive statistics using the central tendency measure and dispersion measure were used to describe the variables of the study. Frequencies and percentages were also used for the categorical variables. T-test, ANOVA, Kruskal–Wallis, and Pearson r tests have been used to examine the differences and relationships, as appropriate. To answer the primary research question regarding the risk and probability of ACEs due to health and risk behaviors, direct binary logistic regression was conducted. A dummy variable was created for the total ACEs score using a cutoff score of 4, as has been conducted in other previous studies [10]. In the analysis of this study, we used 3 or fewer ACEs vs. 4 or more ACEs. When using logistic regression, an odds ratio as small as 1.84 (or 0.5435), with an adjustment for, at most, R^2^ = 0.46, can be detected with 0.80 power at a significance level of 0.05 (two-tailed), from 338 subjects. A confidence interval of 95% was calculated to make the decision to accept the association as statistically significant, based on the criterion that the confidence interval should not exceed 2.01. To test for the underlining assumption, logits were examined for linearity. The examination of plotting the logits versus predictors showed that the curves were linear. Examinations for outliers and interactions (interaction between the main predictors and the outcomes) were conducted. In the backward and forward variable selection of the binary responses, all possible first-order interactions were examined, controlling for the following confounders: age, gender, marital status, educational level, and occupation. The analysis showed that no first-order interactions remained in the final models. In other words, all of the interactions were dropped from the models, and appeared not to be significant. For the predictability and reliability of the model, all variables were included in the main effect model.

## 3. Results

### 3.1. Demographics

A total of 338 individuals completed the survey. The mean age was 26.3 (SD = 8.9), ranging from 18 to 70 years (50% were 18 to 23 years old). Of the participants, 63% were male, 98.8% were Saudi, 74% were unmarried, 48.5% were students, 35.6% were employed, and 49.4% had an undergraduate degree (Table 1).

### 3.2. Adverse Childhood Experiences (ACEs) 

The analysis showed that 68.2% of the respondents to ACEs questions reported at least one ACE, and 34.2% reported four or more ACEs. The highest reported form of ACE was emotional abuse (46.8%), followed by physical abuse (39.3%), and the least reported was sexual abuse (16.8%), followed by family dysfunction (35.7%) (Table 2).

### 3.3. Health and Risk Behaviors 

The analysis showed that the responses of the participants of the study vary across types of diseases, healthy lifestyles, and behaviors. The analysis showed that none of the participants were diagnosed with cancer, and less than 2% of them were diagnosed with liver disease, venereal diseases, or mental illness. The most reported disease was hypertension (14.2%), followed by chronic respiratory disease (11.8%). Regarding the health behaviors, drinking alcohol and using drugs were reported to be low, at 5.3% and 3.2%, respectively. In addition, 34.9% of the participants were smokers, and 39.3% exercised regularly (Table 3).

### 3.4. The Relationship between ACEs, Health and Risk Behaviors, and Sociodemographic Characteristics

#### 3.4.1. Bivariate Analysis 

To examine the relationship between ACE score and the sociodemographic factors, a number of analyses have been conducted. Using the Pearson r, the analysis showed that there was no significant correlation between total ACEs and age (*p* > 0.05). Using the t-test for two independent samples, the analyses showed that there was no statistically significant difference in ACE scores in relation to participants’ gender, nor in relation to their nationality (*p* > 0.05). Using the ANOVA test, the analyses showed that there was no statistically significant difference in ACE scores in relation to marital status nor to working status (*p* > 0.05), while a statistically significant difference was found in ACE scores in relation to level of education (*F* = 2.85, *p* = 0.045). The post hoc comparison (Tukey test) showed that the significant difference was observed between those with less than a high school education and those with a diploma level of education.

Regarding the relationship between health and risk behaviors and sociodemographic characteristics, a number of analyses were also conducted. Using the point biserial correlation, the analyses showed that age was significantly and positively correlated with diabetes mellitus (*r*_bp_ = 0.14, *p* = 0.008) and hypertension (*r*_bp_= 0.16, *p* = 0.003), while no other health and risk factors were significantly correlated with age (*p* > 0.005). Using the t-test for two independent samples, the analysis showed that there was a statistically significant difference in having venereal disease (*t*, 0.83, *p* = 0.04), having mental illness (*t*, 2.1, *p* = 0.041), and being either a current or former smoker (*t*, 2.7, *p* = 0.008) in relation to participants’ gender, with males’ mean scores being higher than females’ in all aforementioned results. However, no other health or risk factors were significantly correlated with gender (*p* > 0.005). 

Kruskal–Wallis was used to examine the differences in relation to marital status. The analyses showed that there were statistically significant differences in being diagnosed with diabetes mellitus (χ^2^ = 6.94, *p* = 0.031), being diagnosed with hypertension (χ^2^= 6.56, *p* = 0.038), being diagnosed with obesity (χ^2^= 6.71, *p* = 0.035), and being a current or former smoker (χ^2^ = 9.28, *p* = 0.010) in relation to participants’ marital status. Kruskal–Wallis showed that one of the social statuses (married, not married, or divorced/widowed) was dominant over the others. However, the test did not identify where this stochastic dominance occurred, nor for how many pairs of groups stochastic dominance was present. 

Kruskal–Wallis was also used to examine the differences in relation to working status. The analyses showed that there were statistically significant differences in being diagnosed with hypertension (χ^2^= 10.80, *p* = 0.005), and being a current or former smoker (χ^2^= 60.01, *p* = 0.050) in relation to participants’ working status. Kruskal–Wallis showed that one of the social statuses (employed, unemployed, or student) was dominant over the others. However, the test did not identify where this stochastic dominance occurred, nor for how many pairs of groups stochastic dominance was present. Moreover, using the Kruskal–Wallis test, no statistically significant differences were found in any of health or risk behaviors in relation to participants’ educational level (*p* > 0.05).

#### 3.4.2. Adjusted Binary Logistic Regression

The model was developed using standardized, forward, and backward stepwise logistic regression. The omnibus test chi-square model (which measures how well the model performs), the Hosmer and Lemoshow chi-square (which tests the goodness-of-fit of the null hypothesis that the model adequately fits the data), and the Negelkerke R^2^ (which is the pseudo r-square statistic which indicates the variation explained by the model, and the −2 Log likelihood) were used to confirm the most robust findings and to compare the proposed models. 

The result showed that the backward LR model adequately fitted the data (χ^2^ Hosmer Lemoshow = 10.29, df = 8, *p* = 0.245). To examine how well the model performed when variables are excluded from the model, the omnibus test was conducted, and the results showed that the most parsimonious (backward) model had performed well because the change in the level of significance was small. This indicated that exclusion of the variables from the model should be recommended (model χ^2^ = 68.90, df = 10, *p* < 0.001) (Table 4).

The parsimonious (backward) binary logistic model showed that perceived depression had a positive and significant effect in the model (Odds Ratio = 8.269, *p* = 0.002). This result infers that participants who had experienced four or more ACEs were eight times more likely to have a diagnosis with depression than those who experienced three or fewer ACEs. In other words, those who were diagnosed with depression were eight times more likely to have experienced four or more ACEs. Furthermore, although the forward and backward model did not reveal a significant effect on ACEs, the full-model (standardized) did show that anxiety had a significant positive effect in the model (Odds ratio = 4.418, *p* = 0.034). This result indicates that that those who had experienced four or more ACEs were four times more likely to have a diagnosis of anxiety than those who had experienced three or fewer ACEs. In other words, those who were diagnosed with anxiety were four times more likely to have experienced four or more ACEs (Table 4). 

## 4. Discussion

Traumatic experiences and adversities are considered as serious antecedents for several mental and psychosocial disorders [8]. Mental health professionals are challenged with providing quality mental healthcare for those exposed to high levels of distress when mental health access and coverage might not be adequately provided [27]. ACEs and their consequences are amongst those issues that require the attention of researchers and clinicians. 

This study examined the relationship between ACEs, the number of chronic diseases and risk behaviors, and sociodemographic factors. We found that depression and anxiety are associated with ACEs, indicating that those who have reported higher levels of ACEs are more likely to develop depression and anxiety, while none of the medical disorders were found to be associated with ACEs. Such findings need to be interpreted cautiously, as we used direct logistic regression while dichotomizing ACEs into two categories: those who scored four or more and those who scored three or less. Those who had experienced four or more ACEs were eight times as likely to develop depression as those who had experienced three or fewer ACEs. On the other hand, those who had experienced four or more ACEs were four times as likely to develop anxiety than those who had experienced three or fewer ACEs. The connection between depression and anxiety and ACEs was well established in this study. The results of this study, while supporting previous reports that ACEs are associated with psychological disturbances, such as depressive feelings, stress, and anxiety [20], do not support a connection between ACEs and the development of physiological or medical disorders [14]. One explanation for this is related to the age group of this study; the majority of the sample were young people, which resulted in a lack of effect or connection between chronic illness and ACEs. This is expected with less representation for those with chronic illnesses. In addition, the results do not support a connection between ACEs and sociodemographics such as age, gender, or working status. 

The findings of this study can be explained in many ways. First, the sample of this study was drawn from rural Saudi areas, which are characterized by harmony in tribal and cultural backgrounds, indicating that their perceptions of adverse life experiences are dependent upon their understanding of the Saudi culture’s, rather than the global, understanding of the concept. This has been reflected in psychological factors rather than physical ones. Another explanation is related to the nature of understanding of such ACEs in their lives. For example, healthcare access and utilization are largely dependent on the availability of services in remote areas, rather than ease of access and service coverage [28]. Thus, measuring psychological consequences was much more feasible and accessible that defining people with medical disorders who might not yet have been diagnosed due to inadequate service coverage. Similarly to other neighboring countries, mental health is underreported and claiming forms of ACEs is stigmatized in the Saudi Arabia, which might have contributed to underreporting and confusing the relationships between social, personal, and medical factors [29]. 

One significant contribution of this study is that we have used a sample of rural adults in Saudi Arabia, when very few studies have paid attention to such groups in Saudi Arabia previously. What characterizes this group is their tribal life and cultural centricity, which makes them unable to find appropriate means of expressing their unpleasant experiences. This adds to our understanding that such people are also at risk for severe forms of psychological disturbances, such as depression and anxiety, which have not been revealed in these populations before. 

Lifestyle is one fragile component of life that people may use to adapt to their unpleasant experiences. This may take multiple forms, including smoking and increasing food consumption, leading to obesity [30]. This has been found to be associated with further metabolic problems, such diabetes and coronary cardiac diseases (CVD). In our study, we found differences in diabetes mellitus, hypertension, obesity, and being a current or former smoker which were related to ACEs. Such findings do support the notion that people might use ineffective coping mechanisms to adapt to their unpleasant and adverse life experiences, leading to further physiological and psychological disorders. The results provide further evidence in terms of ACEs’ association with chronic health conditions later on life. Ineffective coping and risk behaviors, such as smoking, are among the most commonly used unhealthy behaviors among individual who lack mental health and psychosocial resources, and these behaviors lead to physiological deterioration [31]. The findings of this study sustain the connection between disruptive behaviors among children who lack resilience, which make it difficult for them to get through their stress and manage day-to-day issues, causing further psychological and physical harm [32]. 

ACEs contribute to many negative health outcomes; thus, identifying the prevalence of ACEs among the rural population is essential for future health promotion plans, strategies, and programs. It is also important that chronic diseases and risk behaviors be specifically identified among the rural population in order to prioritize health-related actions. Future research should further investigate ACEs and other determinants of health among the rural population, taking into consideration the inclusion of more diverse participants, such as older adults and those from other rural areas. 

### Limitations and Strengths 

This study has some limitations. First, the study uses a convenience method for sampling, which may result in sampling heterogeneity. An example of this is the participants’ age group, as most of the sample comprised young people. Second, the study was conducted at particular rural areas located in a specific geographic region in the country. These rural areas may not represent all rural areas in Saudi Arabia. However, this study uses a sample of rural adults in a country in which they are underrepresented in studies, and less attention is given to them. This study is among very few studies that have focused on the rural population in Saudi Arabia, and is the first study to investigate ACEs and their relation to health in the country.

## 5. Conclusions

This study draws important conclusions in regard to rural population health in Saudi Arabia. The prevalence of ACEs, health and risk behaviors, and the relationship between them has been provided. The results have established knowledge in terms of occurrence of ACEs among the rural population, as well as their impact on health. However, future research should more deeply investigate ACEs and health among rural populations, and should examine other determinants of health, such as sociodemographic determinants, as well as strategies for accessing and providing healthcare services to those remote populations. Policymakers should also facilitate access to healthcare services for rural people, in addition to public health promotion and disease prevention. Screening for ACEs, for example, is highly recommended in primary healthcare centers, as they are available in all geographic settings, including rural areas. Screening would provide continuous and updated information about ACEs, which would help to guide future health-related actions.

## Figures and Tables

**Table 1 healthcare-10-02502-t001:** Demographic characteristics of participants (N = 338).

Variables	N	%
Gender	338	
Male		63.0
Female		37.0
Nationality	338	
Saudi		98.8
Non-Saudi		1.2
Marital Status	338	
Married		23.1
Not married		74.0
Divorced/widowed		3.0
Working Status	338	
Governmental employee		19.5
Private employee		16.3
Unemployed		15.1
Retired		0.6
Student		48.5
Level of Education	338	
>High school		0.9
High school		37.0
Diploma degree		10.7
Undergraduate level		49.4
Graduate level		2.1

**Table 2 healthcare-10-02502-t002:** Adverse Childhood Experiences among the rural population (N = 338).

Variables	% ^a^	Mean	Std. Dev.
ACEs total score (range, 0–19)	68.2	2.96	3.35
0 ACEs	31.8		
1 ACE	15.9		
2 ACEs	9.7		
3 ACEs	8.4		
4 + ACEs	34.2		
Emotional abuse	46.8		
Physical abuse	39.3		
Sexual abuse	16.8		
Neglect	38.6		
Family dysfunction	35.7		

^a^ = Missing for ACEs variable = 30; the total respondents for ACE questions = 308; valid percentages have been reported in this table.

**Table 3 healthcare-10-02502-t003:** Health and risk behaviors among the rural population (N = 338).

Variables	%	Mean	Std. Dev.
Diabetes	9.2	0.09	0.29
Hypertension	14.2	0.14	0.35
Heart disease	5.2	0.05	0.23
Chronic respiratory disease	11.8	0.12	0.32
Liver disease	0.9	0.01	0.09
Venereal disease	2.1	0.02	0.14
Cancer	0	0.00	0.00
Obesity	8.6	0.09	0.28
Depression	8.9	0.09	0.28
Anxiety	8.3	0.08	0.28
Mental illness	2.1	0.02	0.14
Smoking	34.9	0.35	0.48
Drinking alcohol	5.3	0.05	0.22
Using drugs	3.6	0.04	0.19
Regular exercise (at least three times a week)	39.3	0.39	0.49

**Table 4 healthcare-10-02502-t004:** Adjusted binary logistic regression.

Standardized LR	B	S.E.	Wald	*p*	OR	95% CI. for OR	
						Lower	Upper
Diabetes	0.307	0.574	0.286	0.593	1.360	0.441	4.187
Hypertension	0.162	0.485	0.111	0.739	1.175	0.454	3.042
Heart disease	1.010	0.660	2.339	0.126	2.745	0.752	10.014
Chronic respiratory disease	−0.465	0.498	0.872	0.350	0.628	0.237	1.666
Liver disease	0.872	1.522	0.328	0.567	2.392	0.121	47.238
Venereal disease	0.532	1.046	0.259	0.611	1.703	0.219	13.224
Obesity	−0.559	0.553	1.023	0.312	.572	0.193	1.690
Depression	1.914	0.677	7.980	0.005 **	6.778	1.797	25.571
Anxiety	1.486	0.701	4.487	0.034 *	4.418	1.117	17.466
Mental illness	−0.654	1.310	0.249	0.618	.520	0.040	6.777
Smoking	0.303	0.358	0.718	0.397	1.354	0.671	2.732
Drinking alcohol	0.163	0.906	0.032	0.857	1.177	0.199	6.946
Regular exercise	−0.049	0.288	0.029	0.864	.952	0.542	1.673
Age	0.017	0.024	0.519	0.471	1.017	0.971	1.065
Gender	−0.405	0.350	1.333	0.248	.667	0.336	1.326
Marital status	0.668	0.369	3.279	0.070	1.950	0.946	4.016
Working status	0.103	0.112	0.856	0.355	1.109	0.891	1.380
Level of education	0.157	0.153	1.044	0.307	1.170	0.866	1.580
χ2 model = 73.05 (df = 19, *p* < 0.001) χ2 Hosmer Lemoshow = 8.29 (df = 8, *p* = 0.406) Negelkerke R2= 0.292 −2 Log Likelihood = 322.20							
**Forward LR**	**B**	**S.E.**	**Wald**	** *p* **	**OR**	**95% CI. for OR**	
						**Lower**	**Upper**
Heart disease	1.098	0.564	3.792	0.051	2.999	0.993	9.057
Depression	2.113	0.672	9.876	0.002 **	8.269	2.214	30.878
Anxiety	1.176	0.643	3.346	0.067	3.240	0.919	11.418
Age	0.020	0.022	0.827	0.363	1.021	0.977	1.066
Gender	−0.244	0.286	0.724	0.395	0.784	0.447	1.374
Marital status	0.544	0.349	2.439	0.118	1.724	0.870	3.413
Working status	0.112	0.108	1.077	0.299	1.119	0.905	1.382
Level of education	0.162	0.148	1.200	0.273	1.176	0.880	1.573
χ2 model = 68.90 (df = 10, *p* < 0.001) χ2 Hosmer Lemoshow = 10.29 (df = 8, *p* = 0.245) Negelkerke R2= 0.277 −2 Log Likelihood = 326.55							
**Backward LR**	**B**	**S.E.**	**Wald**	** *p* **	**OR**	**95% CI. for OR**	
						**Lower**	**Upper**
Heart disease	1.098	0.564	3.792	0.051	2.999	0.993	9.057
Depression	2.113	0.672	9.876	0.002 **	8.269	2.214	30.878
Anxiety	1.176	0.643	3.346	0.067	3.240	0.919	11.418
Age	0.020	0.022	0.827	0.363	1.021	0.977	1.066
Gender	−0.244	0.286	0.724	0.395	0.784	0.447	1.374
Marital status	0.544	0.349	2.439	0.118	1.724	0.870	3.413
Working status	0.112	0.108	1.077	0.299	1.119	0.905	1.382
Level of education	0.162	0.148	1.200	0.273	1.176	0.880	1.573
χ^2^ model = 68.90 (df = 10, *p* < 0.001) χ^2^ Hosmer Lemoshow = 10.29 (df = 8, *p* = 0.245) Negelkerke R^2^= 0.277 −2 Log Likelihood = 326.35							

* = *p* < 0.05; ** = *p* < 0.01.

## Data Availability

The data presented in this study are available upon request from the corresponding author. The data are not publicly available due to the privacy of participants.

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
