# Peer review of "Adverse Childhood Experiences and Health in Rural Areas of Riyadh Province in Saudi Arabia: A Cross-Sectional Study"

_healthcare, 2022, doi:10.3390/healthcare10122502_

Round 1
Reviewer 1 Report
The abstract is correct, it contains all relevant research information. The issue of the consequences of ACEs taken up in the article, is very important and my assessment of the topic is high. The research is correlational, conducted using self-report questionnaire methods. Therefore, conclusions drawn from these studies must be carefully formulated. But neither did the authors make a mistake of overinterpretation. The authors also see and describe the limitations of their research. The statistical methods used, especially the regression analysis, allowed for the identification of predictive conclusions.
I agree with the authors that the issue of dependence, the impact of ACEs on physical and mental health has already been quite well described in the literature, while research in the rural population is relatively less. Thus, the reviewed article completes the necessary data.
I have doubts about the method of recruiting people for research. The authors write that "Data collectors prepared the data collection scenes at public gathering areas such as shopping malls and public parks". Does this mean there are shopping malls and public parks in rural areas in Saudi Arabia? This requires an explanation. My second doubt concerns why only students were made aware of the study aims. The authors write (lines 129-130): To maintain full privacy and confidentiality, participating STUDENTS were made aware of the study aims. Since all those in shopping malls or in public parks, who agreed to participate in the study, filled out questionnaires, the subsequent selection of respondents on the basis of demographic data, i.e. place of residence, should be described. This was the first main independent variable. How many people were initially tested, how many of them were residents of rural areas? This requires an explanation. Please define the concept “rural population” in Saudi Arabia.
Additionally, explain why was a dummy variable created for the total ACEs score 3 or less ACEs vs 4 or more ACEs, not 1 ACE vs 2 and more, or 2 or less vs 3 or more.
The discussion was interesting, explaining the results well. The article, after introducing the suggested corrections, is worth publishing.
Author Response
Dear Reviewer,
Thank you for the time taking in reviewing our manuscript, and for the valuable comments provided to us. We have addressed the comments and submitted a revised manuscript to the portal. The revisions are marked up using “Track Changes”. We believe that the comments have substantially enhanced our manuscript. Thank you again.
Please find our response to your specific comments in attachment.
Thank you again,

Reviewer 2 Report
This is an interesting study looking at the association between ACEs and mental/physical health outcomes in rural areas of Saudi Arabia.
I like this m/s, found it very readable overall and I have some queries/ comments, the answers to which I hope could further strengthen the m/s.
1. Into- given the analysis of 4+ ACEs in the study I was surprised the impact of 4+ ACEs isn’t included in the introduction. I think including a line about the cumulative would give stronger justification for looking at ACEs as a binary outcome.
2. L32-35 the line about 3 types of abuse (L32) could make more sense incorporated into list on L34
3. L38 I found this sentence hard to follow. I wonder if it would work better as a sentence about the impact of abuse and neglect, and another about how common ACEs are.
4. L40 ‘has’ should be ‘have’
5. Section 2.Materials and Methods becomes a bit bulletpoint like which is out of keeping with the rest of the ms.
6. L75 Sentence could flow better as ‘to examine the relationship between ACEs and chronic diseases in a rural population…’
7. Section 2.2.3 Health and Risk Behaviors Questions – could the authors include e.gs of diseases assessed by the measure?
8. More of a curiosity than to ass in, but I wondered if the authors looked at categories of ACE’s in relation to illness? I was wondering if cumulative ACE’s from certain categories conveyed more risk than others.
9. L154 ‘another word’ should be ‘other words’
10. Given that in L61, the authors mention rural areas having low economic resources, I wondered if the education level of the sample was what the authors expected from a rural community? How might this affect results? Could this have impacted data collection (e.g. use of tablets reducing access for those with lower education or older adults?)
11. L161 ‘50% of them at age of 18 to 23 years of age’ could be ‘50% were 18 to 23 years old’
12. L163 delete ‘had Moreover, the majority of the sample’
13. L185- re smokers and exercise. Wording is inconsistent here; A large portion were smokers (34.9%), but only 39.3% exercised regularly. Given that a larger proportion exercised this phrasing doesn’t make sense.
14. L201 ‘postdoc’ should be ‘posthoc’
15. L210 ‘being mental illness’ should be ‘having a mental illness’
16. L246-249 and repeats the same thing. Same with L252-255. Would phrasing work better the other way e.g. those with 4+ ACEs were more likely to have a diagnosis of x
17. Can the authors clarify why forwards and backwards regressions were applied? Also the tables reporting these models are identical- is this correct?
18. Section 4. Discussion. ‘life devastating problems’ could be better put
19. L266 I think sociodemographics was missed in study aim
20. L321 – this sentence is unclear.
I hope the authors find these useful in revising their ms.
Author Response

(The authors gave the same response as above.)
